# Increased K_V_2.1 Channel Clustering Underlies the Reduction of Delayed Rectifier K^+^ Currents in Hippocampal Neurons of the Tg2576 Alzheimer’s Disease Mouse

**DOI:** 10.3390/cells11182820

**Published:** 2022-09-09

**Authors:** Ilaria Piccialli, Maria José Sisalli, Valeria de Rosa, Francesca Boscia, Valentina Tedeschi, Agnese Secondo, Anna Pannaccione

**Affiliations:** Division of Pharmacology, Department of Neuroscience, Reproductive and Dentistry Sciences, School of Medicine, Federico II University of Naples, 80131 Napoli, Italy

**Keywords:** Alzheimer’s disease, electrophysiology, delayed rectifier K^+^ currents, K_V_2.1, channel clustering, Tg2576 mouse, hippocampal neurons

## Abstract

Alzheimer’s disease (AD) is a neurodegenerative disorder characterized by the progressive deterioration of cognitive functions. Cortical and hippocampal hyperexcitability intervenes in the pathological derangement of brain activity leading to cognitive decline. As key regulators of neuronal excitability, the voltage-gated K^+^ channels (K_V_) might play a crucial role in the AD pathophysiology. Among them, the K_V_2.1 channel, the main α subunit mediating the delayed rectifier K^+^ currents (I_DR_) and controlling the intrinsic excitability of pyramidal neurons, has been poorly examined in AD. In the present study, we investigated the K_V_2.1 protein expression and activity in hippocampal neurons from the Tg2576 mouse, a widely used transgenic model of AD. To this aim we performed whole-cell patch-clamp recordings, Western blotting, and immunofluorescence analyses. Our Western blotting results reveal that K_V_2.1 was overexpressed in the hippocampus of 3-month-old Tg2576 mice and in primary hippocampal neurons from Tg2576 mouse embryos compared with the WT counterparts. Electrophysiological experiments unveiled that the whole I_DR_ were reduced in the Tg2576 primary neurons compared with the WT neurons, and that this reduction was due to the loss of the K_V_2.1 current component. Moreover, we found that the reduction of the K_V_2.1-mediated currents was due to increased channel clustering, and that glutamate, a stimulus inducing K_V_2.1 declustering, was able to restore the I_DR_ to levels comparable to those of the WT neurons. These findings add new information about the dysregulation of ionic homeostasis in the Tg2576 AD mouse model and identify K_V_2.1 as a possible player in the AD-related alterations of neuronal excitability.

## 1. Introduction

Alzheimer’s disease (AD) is a devastating neurodegenerative disorder representing the most common cause of dementia in elderly population. The AD pathology consists of the progressive deterioration of cognitive functions, resulting from an irreversible neuronal loss and the pathological derangement of brain circuits and networks, among other mechanisms [1,2]. According to a growing amount of evidence, the dysregulation of brain activity observed in AD is correlated with the alteration of neuronal excitability. In particular, network hyperexcitability and hypersynchrony, and epileptiform activity have been observed in AD mouse models [3,4,5] and proposed as crucial factors causing the malfunction of hippocampal circuitry and subsequent memory impairment [6,7]. Moreover, an increased risk of developing seizures has been observed in the AD patients with mutations in presenilin 1, presenilin 2, and amyloid precursor protein (APP), or with APP duplications [8,9]. Of note, numerous studies have reported that the Amyloid-β_1–42_ (Aβ_1–42_) peptide, the main component of both the neuritic plaques and intracellular aggregates found in the brain of AD patients, can trigger neuronal hyperactivity [10,11,12]. Accordingly, hyperactive neurons have been found in the hippocampus of different AD transgenic models overproducing the Aβ_1–42_ peptide [5,10,13].

Potassium (K^+^) channels are the most abundant and diverse channels in the mammalian brain. Mediating K^+^ fluxes across the plasma membrane (PM), K^+^ channels are crucial determinants of neuronal excitability [14]. Among them, voltage-gated K^+^ channels (K_V_) are the largest family of K^+^ channels expressed in both the central and peripheral nervous system, where they participate in different biological processes, including the regulation of action potential conduction, neurotransmitter release, synaptic plasticity, and apoptosis [15].

The K_V_2.1 channels are the main α subunits mediating the delayed rectifier K^+^ currents (I_DR_) in hippocampal and cortical pyramidal neurons, where they are present as large, high-density clusters on the PM of somata and proximal dendrites [16,17,18,19,20]. Although K_V_2.1 channels do not participate in action potential repolarization due to their slow kinetics, they work as homeostatic suppressors of the intrinsic neuronal excitability by regulating Ca^2+^ influx during periods of repetitive high-frequency firing [21,22,23], hence shaping long-term changes in both synaptic efficacy and neuronal phenotype [24]. Despite being controversial in the literature at first, the localization of the K_V_2.1 channels has also been ascertained at the axon initial segment of cortical and hippocampal pyramidal neurons, where they may regulate both the frequency and back propagation of the axonal action potential [25,26,27]. Gene mutations altering the K_V_2.1 activity have been linked to epilepsy in humans [28,29,30], while K_V_2.1 knockout mice display hyperactivity and deficits in spatial learning [31].

The K_V_2.1 channels have been extensively investigated for their unique subcellular localization and the ability to sense neuronal activity, which in turn regulates their localization and function through the modulation of their phosphorylation state [32,33]. Indeed, neuronal K_V_2.1 channels are extensively modified by phosphorylation as they display numerous phosphorylation sites for different protein kinases in their cytoplasmic domain [34,35,36]. Interestingly, the degree of phosphorylation, which is constitutively high in neurons, crucially affects K_V_2.1 biophysical properties and localization [23,34,37,38,39], and is strictly coupled with channel clustering [32,35,40]. It has been extensively demonstrated that the K_V_2.1 channels residing within clusters, which represent a large percentage of the channels natively expressed on the neuronal PM, are non-conducting [20,41,42].

Although the involvement of K^+^ channels has been poorly examined in the AD pathophysiology, alterations in the K_V_2.1 activity were observed in some AD experimental models. In particular, K_V_2.1 oxidation in 3xTg hippocampal neurons was supposed to correlate with its increased clusterization and functional downregulation, and to promote neuronal hyperexcitability [43,44]. On the other hand, it was also shown that the injection of the Aβ_25–35_ fragments, which are able to mimic the toxicity of the parent, full-length Aβ_1–42_, is sufficient to induce a significant upregulation of the K_V_2.1 protein expression in the rat brain [45]. Since changes in the K_V_2.1 function could be critical for the homeostatic regulation of neuronal excitability, in the present study we investigated any possible modulation of the K_V_2.1 protein expression and activity in the Tg2576 mouse model of AD, which expresses a human APP gene carrying the double Swedish mutation (KM670/671NL) and produces elevated levels of Aβ_1–42_ [46]. To this aim, we performed (1) electrophysiological experiments on Wild-Type (WT) and Tg2576 primary neurons; (2) Western blotting and immunofluorescence analyses on primary neuronal cultures from Tg2576 and WT mouse embryos; and (3) Western blotting and immunohistochemical analyses on the hippocampus and cerebral cortex of 3-month-old Tg2576 and WT littermates.

## 2. Materials and Methods

### 2.1. Animals

All the animals were handled according to the International Guidelines for Animal Research, and the experimental protocols were approved by the Animal Care and Use Committee of the “Federico II” University of Naples. The heterozygous male Tg2576 mice and WT females were purchased from a commercial source (B6; SJLTg(APPSWE) 2576 Kha, model 1349, Taconic Biosciences, Hudson, NY, USA).

### 2.2. Genotyping: PCR Analysis

Genomic DNA from embryonic brain tissues was isolated by salt precipitation as previously described [47]. Briefly, embryonic brain tissues were harvested during cerebral dissection and then thawed and homogenized with the TRI-reagent (Sigma-Aldrich, Milan, Italy). After adding one volume of chloroform to each sample, the DNA was precipitated with 100% ethanol and centrifuged at 4 °C for 5 min at 16,000× *g*. The DNA pellet was dried at room temperature and then re-suspended in Tris-EDTA buffer. The following primers were used to amplify the DNA region with the human APP Swedish mutation on both types of genomic DNA: 5′-CTGACCACTCGACCAGGTTCTGGGT-3′ and 5′-GTGGAT AACCCCTCCCCC AGCCTAGACCA-3′ (Eurofins Genomics, Ebersberg, Germany). The DNA was amplified to detect the transgenic genotype as previously described [13].

### 2.3. Primary Hippocampal Neurons

Primary neuronal cultures were prepared from the hippocampi of embryonic day-15 WT and Tg2576 mice as described by Ciccone et al. [13].The cells were plated on 35 mm culture dishes coated with poly(D)-lysine hydrobromide with a molecular weight >300,000 (Sigma-Aldrich, Milan, Italy) or onto 25 mm glass coverslips (Glaswarenfabrik Karl Hecht KG, Sondheim, Germany), coated with 100 μg/mL poly(D)-lysine hydrobromide with a molecular weight of 30,000–70,000 (Sigma-Aldrich, Milan, Italy) at a density of one embryo hippocampus/1 mL. Cytosine β-D-arabinofuranoside (10 μM, Sigma-Aldrich, Milan, Italy) was added 3 days after plating to inhibit non-neuronal cell growth. The neurons were cultured at 37 °C in a humidified 5% CO_2_ atmosphere. The experiments were performed between 10–12 days *in vitro* (DIV).

### 2.4. Western Blotting

Total lysates for the immunoblotting analyses were obtained as follows: the primary hippocampal neurons were washed in phosphate buffered saline (PBS) and collected by gentle scraping in ice-cold RIPA buffer containing (in mM): 50 Tris pH 7.4, 100 NaCl, 1 EGTA, 1 PMSF, 1 Na_3_VO_4_, 1 NaF, 0.5% NP-40, and 0.2% SDS, and supplemented with protease inhibitor cocktail II (Roche Diagnostic, Monza, Italy). The nitrocellulose membranes were incubated with the following antibodies: rabbit K_V_2.1 antibody (1:1000, Alomone Labs, Jerusalem, Israel), mouse α-tubulin antibody (1:5000, Sigma-Aldrich, Milan, Italy), and anti-β-actin peroxidase (1:10,000, Sigma-Aldrich, Milan, Italy). The immunoreactive bands were detected with the chemiluminescence system (GE Healthcare, Milan, Italy). The films were developed with a standard photographic procedure, and the quantitative analysis of the bands detected was carried out by densitometric scanning.

### 2.5. Electrophysiological Recordings

K^+^ currents were recorded from WT and Tg2576 primary hippocampal neurons with the patch-clamp technique in whole-cell configuration using a commercially available amplifier Axopatch200B and Digidata1322A interface (Molecular Devices, San Jose, CA, USA), as previously described [48,49]. In most experiments, the whole-cell configuration of the patch-clamp technique was adopted using glass micropipettes of 2.5–5 MΩ resistance. Currents were filtered at 5 kHz and data were acquired and analyzed using the pClamp software (version 9.0, Molecular Devices, CA, USA). The pipette solution contained (in mM): 140 KCl, 2 MgCl_2_, 10 HEPES, 10 glucose, 10 EGTA, and 1 Mg-ATP adjusted at pH 7.4 with KOH. The extracellular solution contained (in mM): 150 NaCl, 5.4 KCl, 3 CaCl_2_, 1 MgCl_2_, 10 HEPES, and adjusted pH 7.4 with NaOH. A total of 50 nM tetrodotoxin (TTX) and 10 μM nimodipine were added to Ringer’s solution to abolish TTX-sensitive Na^+^, and L-type Ca^2+^ current. Monoclonal K_V_2.1 antibody (10 µg mL^−1^; Neuromab, Davis, CA, USA) was added to the pipette solution to block K_V_2.1 currents [22]. The total K^+^ currents (I_K_) are composed of the fast-inactivating (I_A_) currents and the delayed currents (I_DR_). To discriminate K^+^ current components, I_K_= I_A_ + I_DR_, appropriate electrophysiological protocols were used [48,49]. In particular, I_DR_ were isolated by stepping from −80 mV to +40 mV for 250 ms after conditioning pulses at −40 mV, lasting 1.5 s to fully inactivate I_A_. Current amplitudes were measured for I_DR_ at +40 mV at the end of the depolarizing pulse. The steady-state properties of I_DR_ activation were measured as previously described [49]. The current amplitudes recorded upon depolarizing steps from −100 mV to +40 mV (10 mV) were normalized to peak currents at +40 mV and then fitted to the following form of the Boltzmann equation: *G*/*G*_max_ = 1/(1 + exp[(*V*−*V*_1/2_)/*k*]), where *V* is the pulse voltage, *V*_1/2_ is the half-inactivation voltage, and *k* is the slope factor. Possible changes in cell size were taken into account by measuring, in each cell, the membrane capacitance, which is directly related to membrane surface area, and by expressing the current amplitude data as current densities (pA/pF). Capacitive currents were elicited by 5 mV depolarizing pulses from −80 mV and acquired at a sampling rate of 50 kHz. The capacitance of the membrane was calculated according to the following equation: C_m_ = τ_c_ · I_o_/ΔE_m_(1 − I_∞_/I_o_), where C_m_ is the membrane capacitance, τ_c_ is the time constant of the membrane capacitance, I_o_ is maximum capacitance current value, ΔE_m_ is the amplitude of the voltage step, and I_∞_ is the amplitude of the steady state current.

### 2.6. Immunocytochemistry

Hippocampal neurons were washed twice in cold 0.01 M of PBS (pH 7.4) and fixed at room temperature in 4% of paraformaldehyde for 20 min. Following three washes in PBS, cells were blocked in PBS containing 3% of BSA for 30 min, and then incubated overnight at 4 °C with the rabbit anti-K_V_2.1 antibody (1:1000, Alomone Labs, Jerusalem, Israel). The control for the specificity of the antibody against K_V_2.1 was performed with its replacement with normal serum as previously described [50]. Then, cells were washed with PBS and incubated with anti-rabbit Cy2-conjugated antibody (1:200; Jackson Immuno Research Laboratories, Inc., West Grove, PA, USA) for 1 h at room temperature under dark conditions. Cells were finally incubated for 5 min with Hoechst 33342. Cover glasses were mounted with a SlowFade Antifade Kit (Molecular Probes, Life Technologies, Milan, Italy) and acquired by a 63× oil immersion objective using a Zeiss inverted 700 confocal microscope.

### 2.7. Immunohistochemistry

Confocal immunofluorescence procedures were performed as described previously [51,52]. In brief, WT and Tg2576 mice were euthanized at 3 months. Anesthesia was induced with 4% sevoflurane in a mixture of 60% N_2_O and 36% O_2_ and maintained during intracardiac perfusion with 2% sevoflurane in a mixture of 60% N_2_O and 38% O_2_. Transcardial perfusion was carried out with 4% paraformaldehyde in PBS. The brains were sectioned coronally (60 μm) on a vibratome. After blocking with Rodent M block (Biocare Medical, Concord, MA, USA), sections were incubated with the following primary antibodies: rabbit polyclonal anti-K_V_1.2 (1:200) and mouse monoclonal anti-NeuN (1:1000, Sigma-Aldrich, Milan, Italy). Then, sections were incubated with corresponding fluorescence-labeled secondary antibodies (Alexa488- or Alexa594-conjugated anti-mouse or anti-rabbit IgGs). DAPI was used to stain the nuclei. Images were observed using a Zeiss LSM 700 laser (Carl Zeiss, Jena (Turingia), Germany) scanning confocal microscope. Single images were taken with an optical thickness of 0.7 μm and a resolution of 1024 × 1024.

### 2.8. Statistical Analyses

Data are expressed as mean ± standard error of the mean (SEM). Statistical analysis was performed with unpaired *t*-test or one-way analysis of variance followed by the Newman–Keuls test.

## 3. Results

### 3.1. The K_V_2.1 Protein Expression Is Increased in the Hippocampus of 3-Month-Old Tg2576 Mice and in Tg2576 Primary Hippocampal Neurons

Starting from the previous report demonstrating that the Aβ_25–35_ fragments, which are able to mimic the full-length Aβ_1–42_, induced the overexpression of K_V_2.1 in the rat brain [45], we tested the hypothesis that the K_V_2.1 protein expression could be modulated in the Tg2576 mouse, which overproduces the Aβ_1–42_ peptide. To this aim, we performed Western blotting and immunohistochemical analyses on the hippocampus of 3-month-old Tg2576 mice, which start to display alterations in neuronal excitability and early cognitive deficits by this age. The densitometric analysis of Western blotting revealed that the hippocampus of 3-month-old Tg2576 mice displayed higher levels of K_V_2.1 protein expression than the WT littermates (Figure 1A). Accordingly, confocal double immunofluorescence analyses showed that the CA1 pyramidal neurons of the Tg2576 mice exhibited a more pronounced K_V_2.1 immunofluorescence in comparison with the WT littermates (Figure 1B). In line with previous evidence [22,40,53], the double confocal images also revealed the typical clustered K_V_2.1 immunostaining on the PM of the hippocampal CA1 pyramidal neurons of both the WT and Tg2576 mice. Based on the assumption that primary neuronal cultures from Tg2576 mouse embryos, which accumulate Aβ_1–42_ with the time in culture, recapitulate some of the main features of the Aβ_1–42_ neurotoxicity [54], we also performed Western blotting experiments on hippocampal neurons cultured from Tg2576 and WT mouse embryos. Interestingly, we found that the Tg2576 primary hippocampal neurons also exhibited an increase in the K_V_2.1 protein expression in comparison with the WT neurons (Figure 1C).

### 3.2. Tg2576 Primary Neurons Display Decreased K_V_2.1-Dependent I_DR_ Density

To test the hypothesis that the K_V_2.1 overexpression could be accompanied by any alterations in the channel activity, we recorded I_DR_ in pyramidal neurons cultured from Tg2576 and WT mouse embryos by means of the whole-cell patch-clamp technique. In order to electrophysiologically isolate the I_DR_ conductance from the total K^+^ currents, we used a previously established protocol [48]. In particular, to record the I_DR_ component, from a holding potential of −80 mV we applied a conditioning pulse at −40 mV lasting 1.5 s to fully inactivate the fast-inactivating current (I_A_) component, followed by 250 ms steps from −80 to +40 mV (Figure 2A, bottom). Of note, the Tg2576 pyramidal neurons displayed a significantly lower I_DR_ density than the WT neurons (Figure 2A–C). Moreover, the analysis of the activation properties of the I_DR_ revealed a −10 mV hyperpolarizing shift of the steady-state activation curve in the Tg2576 pyramidal neurons compared with the WT neurons (Figure 2D).

To investigate whether the reduction of the I_DR_ in the Tg2576 neurons was actually due to the downregulation of the K_V_2.1 currents, we assessed the contribution of the K_V_2.1 component to both the Tg2576 and WT neuronal I_DR_ by recording these currents upon the intracellular diffusion of the anti-K_V_2.1 monoclonal antibody (10 µg mL^−1^) through the recording pipette. Figure 3A shows a representative time-course of the effect of the anti-K_V_2.1 antibody in the WT and Tg2576 neurons. As previously reported [22], the diffusion of the anti-K_V_2.1 monoclonal antibody into the cells drastically reduced the I_DR_ in the WT neurons (Figure 3A,B), hence indicating that the K_V_2.1 currents were the major components of the I_DR_ in these neurons. By contrast, the anti-K_V_2.1 antibody produced a markedly lower effect in reducing the I_DR_ at 6 min in the Tg2576 neurons than in the WT neurons (Figure 3A–C), hence suggesting that the K_V_2.1 current component in the Tg2576 neurons was lower than in the WT neurons. The subtraction of the antibody-resistant currents at 6 min from the initial current illustrated in Figure 3C clearly represents this result. The representative traces of the I_DR_ and relative current–voltage relationship recorded by 6 min, without (top) and with (bottom) the anti-K_V_2.1 monoclonal antibody in the recording pipette, are illustrated in Figure 3D and 3E.

In agreement with the major contribution of the K_V_2.1 component to the I_DR_ in WT neurons, we found that the steady-state parameters of the I_DR_ activation recorded in the WT neurons upon the intracellular diffusion of the anti-K_V_2.1 antibody were significantly different from those recorded in the WT neurons in the absence of the antibody (Figure 3F). This data likely suggested that upon the blockade of the K_V_2.1 currents by the anti-K_V_2.1 antibody, the contribution of non-K_V_2.1 components characterized by different activation properties were unmasked. Interestingly, the left-shift of the voltage dependence of the steady-state activation curve of the I_DR_ observed in the WT neurons upon the K_V_2.1 blockade was similar to that observed in the Tg2576 neurons (Figure 3F), thereby suggesting that a loss of the K_V_2.1 currents might underlie the hyperpolarizing shift of the steady-state activation curve observed in the Tg2576 neurons.

**Figure 3 cells-11-02820-f003:**
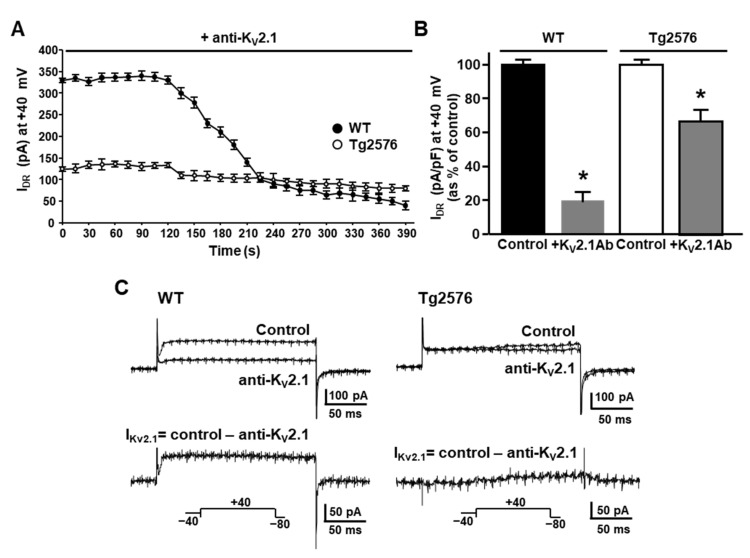
Effect of the intracellular diffusion of the anti-K_V_2.1 monoclonal antibody on I_DR_ recorded in the WT and Tg2576 primary hippocampal neurons. (**A**) Time-dependent inhibitory effect of the anti-K_V_2.1 monoclonal antibody (10 µg mL**^−^**^1^) on I_DR_ recorded in WT and Tg2576 primary hippocampal pyramidal neurons. (**B**) Quantification of I_DR_ densities, at +40 mV, before (Control) and after 6 min of the anti-K_V_2.1 intracellular diffusion (+K_V_2.1Ab). Data are expressed as percentage of WT and Tg2576 internal controls. Values are expressed as mean ± SEM of 3 independent experimental sessions. * *p* < 0.001 vs. internal controls. (**C**) Superimposed representative traces of I_DR_ at +40 mV recorded before and after 6 min of intracellular diffusion of the anti-K_V_2.1 antibody (**top**), and representative traces resulting from the subtraction of the currents resistant to the anti-K_V_2.1 antibody from the initial currents (**bottom**) in WT (**left**) and Tg2576 (**right**) neurons. (**D**) Representative traces of I_DR_ recorded in control conditions and upon 6 min of intracellular diffusion of the anti-K_V_2.1 monoclonal antibody (10 µg mL**^−^**^1^) in WT (**left**) and Tg2576 (**right**) primary hippocampal pyramidal neurons. Protocols are shown in the lower part of the panel. (**E**) I/V relationships for the I_DR_ recorded in control conditions and upon 6 min of intracellular diffusion of the anti-K_V_2.1 antibody in the WT (**left**) and Tg2576 primary hippocampal neurons (**right**). (**F**) Steady-state activation curves of I_DR_ recorded in WT and Tg2576 hippocampal neurons in control conditions and in WT neurons upon 6 min of intracellular diffusion of the anti-K_V_2.1 antibody. Values are expressed as mean ± SEM of 3 independent experimental sessions. * *p* < 0.001 vs. WT.

### 3.3. Glutamate Recovers the Reduction of the K_V_2.1-Mediated I_DR_ Caused by Increased Channel Clustering

As mentioned above, the majority of the K_V_2.1 channels expressed in pyramidal neurons reside within high-density cell surface clusters. Clustered channels are held in a non-conducting state, while non-clustered channels, which are spread diffusely over the cell surface, mediate the majority of the whole I_DR_. Based on this aspect, we hypothesized that the reduction of the K_V_2.1-mediated I_DR_ in the Tg2576 neurons could be due to an increase in channel clustering. Further prompted by the evidence that the overexpression of K_V_2.1 results in increased channel clustering [55], we performed confocal experiments to assess possible changes in channel clustering in the Tg2576 neurons. To this aim, both the Tg2576 and WT hippocampal neurons were fixed and stained with the anti-K_V_2.1 rabbit polyclonal antibody (1:1000, Alomone Labs, Jerusalem, Israel) and subjected to imaging on a confocal microscope. Interestingly, if compared with the WT neurons, the Tg2576 neurons displayed an evident increase in channel clustering. In particular, while microclusters of K_V_2.1 channels were present in both the WT and Tg2576 neurons, markedly larger clusters of K_V_2.1, usually defined as “macroclusters”, were clearly detectable only in the Tg2576 neurons (Figure 4A).

To functionally investigate whether the increase of channel clustering was actually responsible for the reduction of the K_V_2.1 currents previously observed in the Tg2576 neurons, we tested the effect of a declustering stimulus on the K_V_2.1 currents in both the WT and Tg2576 neuronal cultures. In particular, based on the evidence provided by the Trimmer group that glutamate is able to induce a calcineurin-dependent K_V_2.1 dephosphorylation and the subsequent dispersal of channel clusters [40], we exposed both the WT and Tg2576 neurons to glutamate (10 μM, 10 min). In line with the previous results provided by Mohapatra and coworkers [33], we found that the exposure to 10 μM glutamate for 10 min significantly increased the I_DR_ in the WT neurons (Figure 4B–E). Notably, glutamate exerted a stimulatory effect on the I_DR_ also in the Tg2576 neurons, restoring the I_DR_ density at 10 min to levels similar to those recorded in the WT neurons (Figure 4B–E). Consistent with the higher levels of K_V_2.1 channels expressed in the Tg2576 neurons and become available for conduction upon the glutamate-induced declustering, the stimulatory effect of glutamate on the I_DR_ seemed to be greater in the Tg2576 neurons than in the WT neurons (Figure 4B,C). Interestingly, we also found that the steady-state parameters of the I_DR_ activation in the Tg2576 neurons upon glutamate exposure were significantly different if compared with those in the Tg2576 neurons under control conditions (Figure 4F). Of note, the shift in the voltage-dependence of activation induced by glutamate restored the steady-state activation curve to values comparable to those of the WT neurons (Figure 4F).

To exclude the possibility that the stimulatory effect of glutamate on the I_DR_ could be unrelated to K_V_2.1 declustering, we exposed both the WT and Tg2576 neurons to glutamate (10 μM, 10 min) in the presence of the anti-K_V_2.1 antibody in the recording pipette. Remarkably, when K_V_2.1 was blocked by the intracellular diffusion of the anti-K_V_2.1 antibody, the glutamate-induced upregulation of the I_DR_ was not observed in the WT or in the Tg2576 neurons (Figure 5A,B). This result clearly shows that the upregulation of the I_DR_ induced by glutamate was mediated by the specific upregulation of the K_V_2.1 currents resulting from the glutamate-induced channel declustering.

### 3.4. K_V_2.1 Clustering Is Increased in Cortical Layer V Pyramidal Neurons of the Somatosensory Cortex of 3-Month-Old Tg2576 Mice

To investigate whether K_V_2.1 clustering was increased in the Tg2576 model also *in vivo*, we performed confocal double immunofluorescence analyses on the somatosensory cortex of both 3-month-old WT and Tg2576 mice, where the K_V_2.1 protein expression and function have been extensively characterized [23,56]. Interestingly, the confocal double immunofluorescence analysis revealed that the anti-K_V_2.1 antibody depicted a clear and pronounced clustered PM labeling of NeuN-positive neurons in layer V of the somatosensory cortex of both 3-month-old WT and Tg2576 mice (Figure 6A). More important, quantitative analyses showed a significantly higher number of cortical neurons displaying K_V_2.1 clustering in the layer V of the Tg2576 mice in comparison with the WT littermates (Figure 6B).

## 4. Discussion

In the present study, we demonstrated that K_V_2.1 protein expression was significantly increased in primary neurons from the Tg2576 mouse, which, carrying the human Swedish double mutation of the APP, accumulates the Aβ_1–42_ peptide overtime in culture [54,57]. However, as a consequence of the increased channel clustering, the K_V_2.1-mediated currents were markedly reduced in the Tg2576 neurons, resulting in a decrease in the whole I_DR_ along with clear changes in their voltage-dependence of activation. Moreover, we showed that the K_V_2.1 expression and clustering levels were modulated also in vivo in Tg2576 mice.

First, Western blotting analyses on the hippocampal lysates from 3-month-old Tg2576 mice revealed a significant increase in the K_V_2.1 protein levels in comparison with the WT hippocampal lysates, hence providing the first evidence, to our knowledge, that the K_V_2.1 protein expression is modulated in a transgenic in vivo model of AD. In agreement, immunohistochemical analyses on WT and Tg2576 brain sections showed a more pronounced K_V_2.1 immunostaining in the CA1 pyramidal neurons of the Tg2576 mice in comparison with the WT littermates. Of note, Tg2576 mice, a widely used transgenic model of the AD pathology, start by 3 months of age to intracellularly accumulate Aβ_1–42_ trimers, which have been implicated in the appearance of precocious cognitive deficits and high seizure susceptibility [58,59,60]. In line with the previous evidence demonstrating that the neuronal cultures from Tg2576 mouse embryos recapitulate some of the main features of the Aβ_1–42_ neurotoxicity [54], we found that the K_V_2.1 protein expression was also increased in the hippocampal neurons cultured from Tg2576 embryos.

Based on the evidence that K_V_2.1 is the main channel α subunit mediating I_DR_ in hippocampal neurons [16], we investigated whether its overexpression could affect the whole I_DR_ density in the Tg2576 primary hippocampal neurons. Consistent with the aforementioned literature, our electrophysiological experiments revealed that K_V_2.1 was the main channel α subunit mediating I_DR_ in the WT pyramidal neurons, since the intracellular diffusion of the anti-K_V_2.1 monoclonal antibody induced a significant reduction of the I_DR_ in these neurons. Interestingly, we noticed that the I_DR_ density was significantly reduced in the Tg2576 neurons compared with the WT neurons. More important, the intracellular diffusion of the anti-K_V_2.1 antibody produced a markedly lower effect in promoting the decrease of the I_DR_ density in the Tg2576 neurons than in the WT neurons, hence suggesting that the percentage of the conducting K_V_2.1 channels in the Tg2576 neurons was reduced or functionally compromised. In addition, the analysis of the voltage-dependence of activation revealed significant differences in the steady-state parameters of the I_DR_ activation in the Tg2576 neurons compared with the WT neurons. Interestingly, the shift of the voltage-dependence of activation to more negative values observed in the Tg2576 neurons was similar to that produced by the intracellular diffusion of the anti-K_V_2.1 antibody in the WT neurons. This observation suggested that the left-shift in the voltage-dependence of activation occurring in the Tg2576 neurons could reflect a lower contribution of the K_V_2.1 current component to the whole I_DR_.

Although apparently divergent from the higher expression levels observed in the Tg2576 neurons, our functional results are in line with previous evidence clearly indicating that the K_V_2.1 channel conductance decreases as the expression density on the cell surface membrane increases. Indeed, the Tamkun group provided solid data demonstrating that the percentage of non-conducting K_V_2.1 channels strictly depends on their expression levels and consequent channel density, whereas it appeared to be independent on cluster localization *per se* [61]. On the other hand, it has been extensively reported that the vast majority of the K_V_2.1 channels residing within clusters are held in a non-conducting state [20,41,42], in agreement with the fact that the high concentration of channels within a cluster results in a 5-fold to 10-fold greater density in this restricted space than outside [61]. Prompted by the evidence that the overexpression of K_V_2.1 promotes its clusterization, probably by inducing the fusion of smaller clusters to form macroclusters [55], we also hypothesized that the lower K_V_2.1-mediated I_DR_ density observed in the Tg2576 neurons could be due to an increase in channel clustering. In confirmation of this hypothesis, immunocytochemical analyses showed a marked change in the K_V_2.1 clustering in the Tg2576 pyramidal neurons. In particular, the K_V_2.1 immunostaining revealed that, along with the typical clustered expression detected in both the WT and Tg2576 neurons, a massive presence of macroclusters was detectable only in the Tg2576 neurons.

While the causal relationship between the K_V_2.1 overexpression and its clustering has not been fully defined, it was instead clearly demonstrated that K_V_2.1 channel clustering strictly depends on its phosphorylation state. Indeed, it was reported that the phosphorylation of specific residues of K_V_2.1, identified in mutagenesis studies, drastically affects both the clustered localization and the voltage-dependent gating of the channel [32,34,35,40,62]. Intriguingly, the phosphorylation of the proximal restriction and clustering domain is propaedeutic to the binding of K_V_2.1 to the endoplasmic reticulum (ER)-resident VAPA and VAPB (vesicle-associated membrane and protein-associated protein A and B) and the subsequent formation of specialized ER–PM junctions. The function of the K_V_2.1-induced ER–PM junctions appeared to be unrelated to K^+^ conductance, hence providing an explanation for the non-conducting state of the K_V_2.1 channels residing within clusters [20,42,63,64,65]. Physiologically, K_V_2.1 is highly phosphorylated in resting neurons and undergoes a neuronal activity-induced dephosphorylation through a Ca^2+^/calcineurin-dependent mechanism [32,40]. Consistently, stimuli that increase neuronal activity and/or cytoplasmic Ca^2+^ levels induce the dispersal of K_V_2.1 clusters and enhance the voltage-sensitivity [32,35,40].

Therefore, we tested the possibility that a declustering stimulus could restore the K_V_2.1 currents in the Tg2576 neurons. In particular, based on the results provided by the Trimmer group demonstrating that glutamate stimulation rapidly induces a calcineurin-dependent dephosphorylation of K_V_2.1 and its subsequent translocation from clusters to a more diffuse localization [40,63], we measured the I_DR_ in the Tg2576 neurons upon glutamate application. Interestingly, we found that a brief application of 10 µM glutamate was actually able to increase the I_DR_ in the Tg2576 neurons. In line with the results provided by the Trimmer group [33,35,66], the same treatment induced an increase in I_DR_ also in the WT neurons, where the majority of K_V_2.1 channels are physiologically clustered. However, the increase of the I_DR_ induced by glutamate exposure in the Tg2576 neurons was greater than that observed in the WT neurons, arguably due to the larger amount of clustered K_V_2.1 channels present in the Tg2576 neurons and become available for conduction upon glutamate-induced declustering. In this regard, it is worth noting that, unlike the results provided by Misonou and co-workers, a study by the Tamkun group showed that the K_V_2.1 declustering induced by swinolide A or alkaline phosphatase resulted in no change in current density or voltage-dependence of the channel [41]. Nonetheless, the same authors stated that it cannot be ruled out that distinct declustering stimuli may have different results or that the cellular model they used in their study, namely HEK cells, does not completely mimic a neuronal model.

To exclude any possible non-specific effects of glutamate on I_DR_, we recorded the I_DR_ in both the WT and Tg2576 neurons upon glutamate exposure in the presence of the anti-K_V_2.1 antibody in the recording pipette. Importantly, when blocking K_V_2.1 with the anti-K_V_2.1 antibody, the glutamate-induced upregulation of the I_DR_ was not observed in the WT or in the Tg2576 neurons, hence indicating that the positive effect of glutamate on the I_DR_ density was related to K_V_2.1 declustering. Interestingly, glutamate application also produced a significant depolarizing shift in the voltage-dependence of activation in the Tg2576 neurons. Of note, this depolarizing shift restored the conductance-voltage relationship to values near to those of the WT neurons. In this respect, it should be noted that molecular studies by the Trimmer group clearly demonstrated that the calcineurin-dependent dephosphorylation of K_V_2.1 channels yields hyperpolarizing shifts in their voltage-dependent activation, which seems to be not in line with our results [34,35,40]. However, it should be considered that, while the treatment with glutamate made the K_V_2.1 current component newly available, thus significantly increasing its contribution to the whole I_DR_, the I_DR_ registered in the Tg2576 neurons in control conditions had to arguably be attributed to non-K_V_2.1 components with different activation properties. In agreement, we observed that the blockade of K_V_2.1 in the WT neurons produced a left shift in the steady-state I_DR_ activation curve, likely suggesting that the loss of the K_V_2.1 current component brought out a major contribution of non-K_V_2.1 components to the voltage-dependence of activation. Thereby, it is conceivable that the functional effect of K_V_2.1 modulation by glutamate on the I_DR_ activation properties could reflect the restoring of the K_V_2.1 current contribution to the whole I_DR_ in the Tg2576 neurons, with no change in the kinetic properties of the channel. In this respect, it is worth noting that this is the first report, to our knowledge, on the effect of glutamate on the K_V_2.1-mediated I_DR_ in a cellular model where the K_V_2.1 channels are both overexpressed and more clustered.

Of note, we have previously demonstrated that primary hippocampal neurons from the Tg2576 mouse display increased excitability if compared with WT neurons [13]. In the present study, we found that the overexpression of K_V_2.1 and its increased clusterization occurred in Tg2576 pyramidal neurons both *in vitro* and *in vivo*. Our *in vitro* experiments also revealed that the increased K_V_2.1 channel clustering caused a significant reduction of the K_V_2.1-mediated currents and considerable changes in the voltage-dependence of activation of the I_DR_ in Tg2576 neurons. Therefore, it is likely that similar functional changes may occur in the Tg2576 pyramidal neurons *in vivo*. More importantly, a downregulation of the K_V_2.1-mediated currents might be expected to have a crucial impact on hippocampal and cortical excitability in the Tg2576 mouse. In this regard, a previous study by Frazzini and colleagues [43] revealed that the reduction in the K_V_2.1 current density due to channel oxidation was associated with the spontaneous hyperexcitability observed in the neurons from the 3xTg-AD mouse, a transgenic model characterized by elevated levels of oxidative stress in the early phases of the pathology [67]. In particular, they observed increased [Ca^2+^]_i_ spike frequency in the 3xTg-AD neurons, where the K_V_2.1-mediated currents were downregulated, and obtained similar results in the non-transgenic neurons upon the pharmacological blockade of K_V_2.1 [43].

Tg2576 mice have been reported to display an ectopic expression of the Neuropeptide Y, a marker of chronic seizures, and an aberrant network activity even before the onset of memory deficits [60,68,69,70], hence emerging as an animal model particularly suitable to study neuronal hyperactivity in association with Aβ_1–42_ accumulation. However, I_DR_ had not been yet investigated in this transgenic model. Nonetheless, in recent studies, the D’Amelio group reported that pyramidal neurons of both dorsal and ventral CA1 of the Tg2576 mouse displayed altered firing and synaptic transmission in the pre-plaque stages of AD [59,71,72]. Intriguingly, based on the results of computational models, they suggested that the persistent perturbation of the I_DR_ conductance and kinetic properties could be a critical determinant of the firing alterations observed in the Tg2576 ventral CA1 neurons [72].

Finally, considered the non-conducting and structural functions of K_V_2.1, further implications unrelated to neuronal excitability could arise from its overexpression in AD. Indeed, immediately facing astrocytic processes and lying on neuronal cell membranes over the ER subsurface cisternae [73], the K_V_2.1 clusters play a key role in inducing ER-PM junctions and provide specialized molecular platforms to regulate both somatodendritic Ca^2+^ signaling and ion channel trafficking. In particular, it has recently been shown that K_V_2.1 clustering at neuronal ER-PM junctions promotes the clustering of PM L-type Ca^2+^ channels and their coupling to ryanodine receptor ER Ca^2+^ release channels [74], hence allowing the generation of localized Ca^2+^ release events. Moreover, the fact that K_V_2.1 clusters induce the formation of -and localize at- ER-PM junctions confers to K_V_2.1 the ability not only to trigger a structural remodeling of the ER but also to affect neuronal Ca^2+^ handling by modulating ER Ca^2+^ uptake [75]. Of note, we have recently reported about the occurrence of an ER Ca^2+^ remodeling in the Tg2576 primary neurons. In particular, we showed that the enhanced activity of the Na^+^/Ca^2+^ exchanger 3 promotes an increased ER Ca^2+^ refilling in Tg2576 neurons in comparison with WT neurons [47]. Moreover, we demonstrated that the I_A_-mediating K_V_3.4 channel subunit, which is overexpressed and functionally upregulated in Tg2576 reactive astrocytes, regulates astrocytic [Ca^2+^]_i_ transients and ER Ca^2+^ levels [52,76]. The findings reported in the present study bring further information about the ionic dysregulation occurring in the Tg2576 mouse brain, hence suggesting new pathways to explore in studying the alterations of Ca^2+^ signaling and homeostasis in AD and the involvement of potassium channels in these processes.

## 5. Conclusions

Undoubtedly, the involvement of I_DR,_ and in particular of the K_V_2.1 currents, as well as of many other ionic mechanisms, in the AD-related alterations of neuronal excitability deserves much attention. In fact, how AD brains develop hyperexcitability is still largely unknown. Changes in the excitatory/inhibitory balance [77,78,79] and the increase in intrinsic membrane excitability properties [80] have been suggested as possible explanations, with the Aβ peptide proposed as the main determining factor. In this context, the Aβ-induced alteration of protein expression and function of different K^+^ channels could represent a crucial mechanism promoting hyperexcitability and altered neuronal activity. Considering the pivotal role of the K_V_2.1-mediated I_DR_ in regulating the intrinsic excitability of pyramidal neurons, further studies should be performed to address this issue in AD models. In addition, since the alteration of cytosolic and organellar Ca^2+^ homeostasis is considered a critical event in the AD pathogenesis [81], the impact of the K_V_2.1 modulation on these processes should be further investigated in AD models.

## Figures and Tables

**Figure 1 cells-11-02820-f001:**
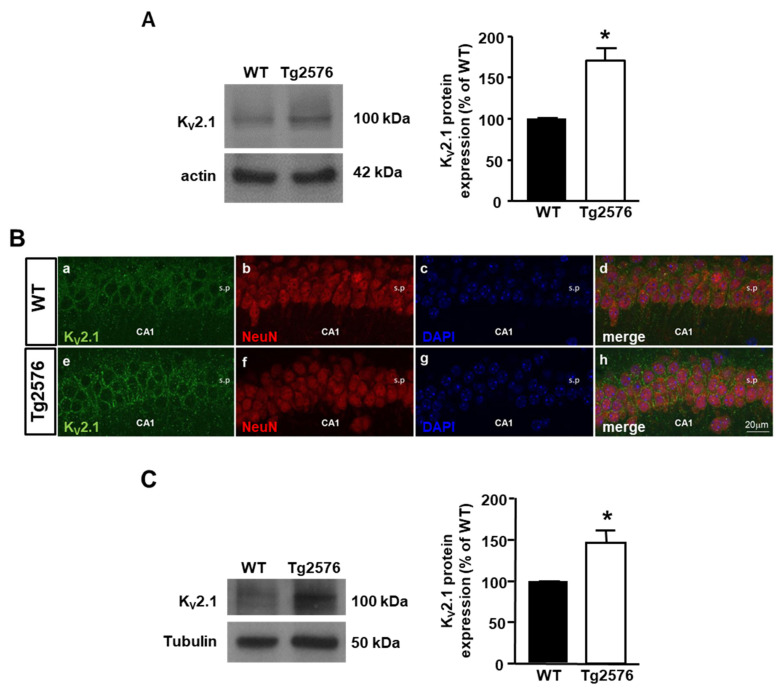
K_V_2.1 protein expression in the hippocampus of 3-month-old Tg2576 mice and in Tg2576 primary hippocampal neurons. (**A**) Representative Western blotting (**left**) and densitometric quantification (**right**) of K_V_2.1 protein expression in the hippocampus of 3-month-old WT and Tg2576 mice. (**B**) Representative confocal double immunofluorescence images displaying K_V_2.1 (green), NeuN (red), and DAPI (blue) distribution in the pyramidal layer of the CA1 hippocampal subfield of WT (**a**–**d**) and Tg2576 (**e**–**h**) mice. Scale bar: 20 µm. (**C**) Representative Western blotting (**left**) and densitometric quantification (**right**) of K_V_2.1 protein expression in WT and Tg2576 primary hippocampal neurons after 12 DIV. Values are expressed as mean ± SEM of 3 independent experimental sessions (* *p* < 0.01 vs. WT).

**Figure 2 cells-11-02820-f002:**
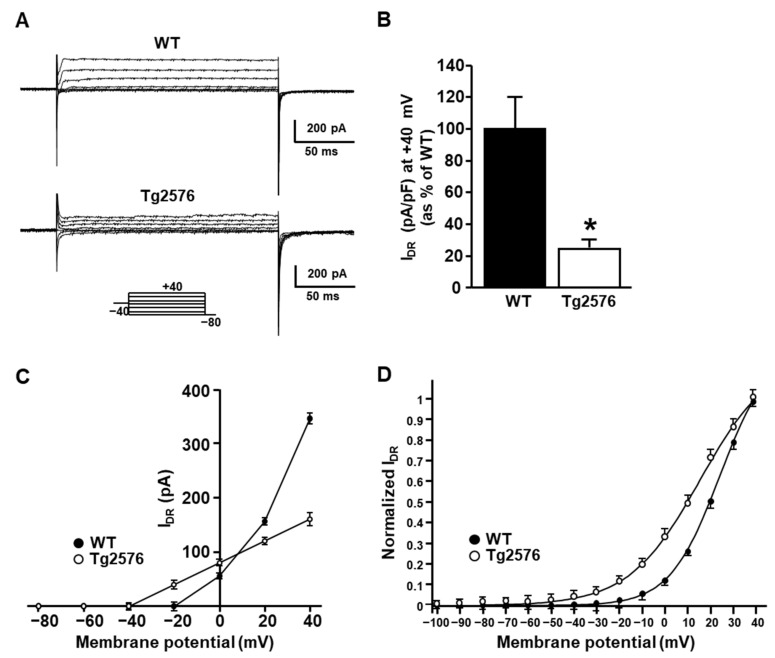
Delayed rectifier K^+^ currents in Tg2576 primary hippocampal pyramidal neurons. (**A**) Representative traces of I_DR_ recorded in the WT (**top**) and Tg2576 (**bottom**) primary hippocampal pyramidal neurons at 12 DIV using the voltage protocol shown in the lower part of the panel. (**B**) Quantification of I_DR_ densities represented in panel *A*. The peak value I_DR_ was measured at the end of the +40 mV depolarizing pulse. Values are expressed as mean ± SEM of 3 independent experimental sessions. * *p* < 0.001 vs. WT. (**C**) Current-voltage (I/V) relationships for the I_DR_ recorded in the WT and Tg2576 hippocampal neurons. Values are expressed as mean ± SEM of 3 independent experimental sessions. (**D**) Steady-state activation properties of I_DR_ recorded in WT and Tg2576 primary hippocampal neurons. Values are expressed as mean ± SEM of 3 independent experimental sessions. * *p* < 0.001 vs. WT.

**Figure 4 cells-11-02820-f004:**
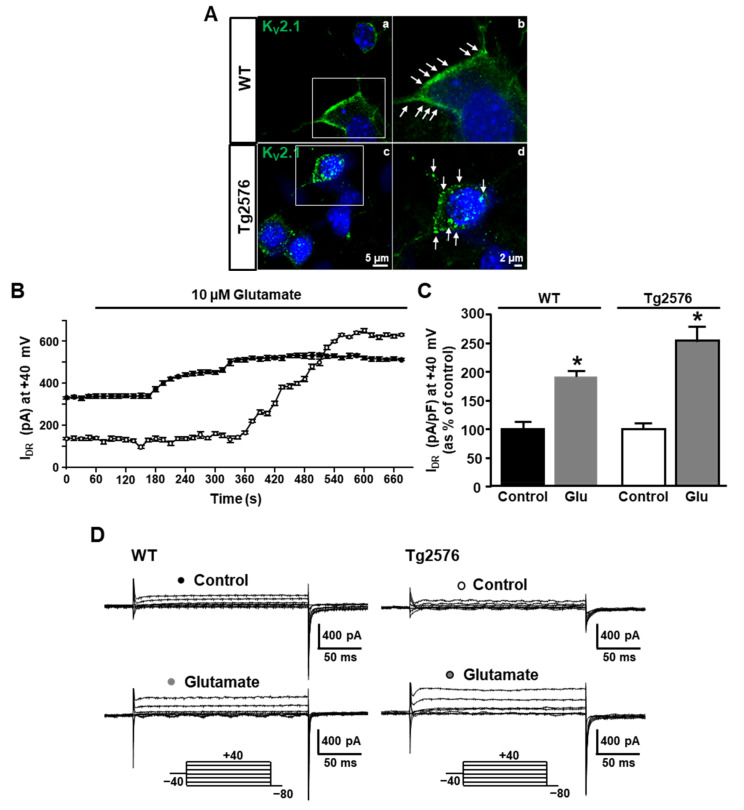
K_V_2.1 clustering and effect of glutamate on I_DR_ in WT and Tg2576 primary hippocampal pyramidal neurons. (**A**) Representative confocal images of pyramidal neurons from Tg2576 (**c**,**d**) and WT (**a**,**b**) primary hippocampal cultures stained with anti-K_V_2.1 antibody (green). On the right, a magnification of neuronal somata from the confocal images shown on the left. (**B**) Time-dependent stimulatory effect of glutamate (10 µM) on I_DR_ recorded in WT and Tg2576 primary hippocampal pyramidal neurons. (**C**) Quantification of I_DR_ densities, at +40 mV, before (Control) and after 11 min of 10 µM glutamate exposure (Glu). Data are represented as percentage of WT and Tg2576 internal controls. Values are expressed as mean ± SEM of 3 independent experimental sessions. * *p* < 0.001 vs. internal controls. (**D**) Representative traces of I_DR_ recorded in control conditions and upon 11 min of 10 µM glutamate exposure in WT (**left**) and Tg2576 (**right**) primary hippocampal pyramidal neurons. Protocols are shown in the lower part of the panel. (**E**) I/V relationships for the I_DR_ recorded in control conditions and upon 11 min of 10 µM glutamate exposure in the WT (**left**) and Tg2576 primary hippocampal neurons (**right**). (**F**) Steady-state activation curves of I_DR_ recorded in WT and Tg2576 hippocampal neurons in control conditions and in Tg2576 neurons upon 11 min of 10 µM glutamate exposure. Values are expressed as mean ± SEM of 3 independent experimental sessions.

**Figure 5 cells-11-02820-f005:**
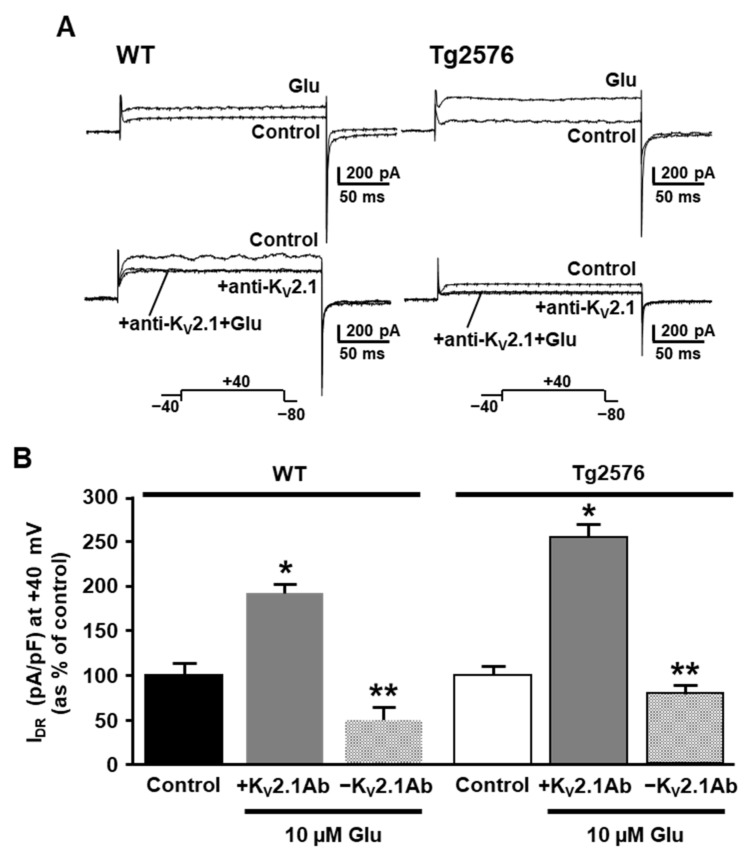
Effect of the anti-K_V_2.1 monoclonal antibody on the positive modulation of I_DR_ by glutamate in WT and Tg2576 primary hippocampal neurons. (**A**) Superimposed representative traces of I_DR_ at +40 mV recorded before (Control) and after 11 min of 10 µM glutamate (Glu), and representative traces recorded before and after the intracellular diffusion of the anti-K_V_2.1 antibody and after 11 min of 10 µM glutamate exposure in the presence of the anti-K_V_2.1 antibody. (**B**) Quantification of I_DR_ densities, at +40 mV, in control conditions and upon 11 min of 10 µM glutamate exposure with (+K_V_2.1Ab) and without the anti-K_V_2.1 antibody (−K_V_2.1Ab) in the recording pipette. Data are expressed as percentage of WT and Tg2576 internal controls. Values are expressed as mean ± SEM of 3 independent experimental sessions. * *p* < 0.001 vs. control; ** *p* < 0.001 vs. GLU + K_V_2.1Ab.

**Figure 6 cells-11-02820-f006:**
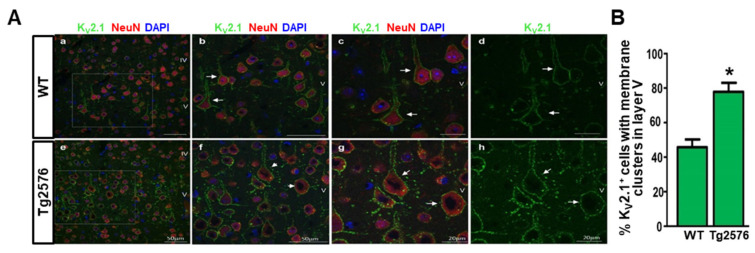
Analysis of K_V_2.1 clustering in the layers IV–V of the somatosensory cortex of 3-month-old WT and Tg2576 mice. (**A**) Representative confocal double immunofluorescence images displaying K_V_2.1 (green), NeuN (red), and DAPI (blue) distribution in the layers IV–V of the somatosensory cortex of WT (**a**–**d**) and Tg2576 (**e**–**h**) mice. **b** and **f**: higher magnification images of the frames depicted in panel *a* and *e*, respectively. **c**,**d** and **g**,**h**: higher magnification images of K_V_2.1-positive neurons depicted by arrows in panels *b* and *f*, respectively. Scale bars in *a*,*b*,*e*,*f*: 50 μm; in *c*,*d*,*g*,*h*: 20 μm. (**B**) Quantification of K_V_2.1 positive neurons with plasma membrane clusters in the layer V of the somatosensory cortex of WT and Tg2576 mice. Data were normalized to the total number of neuronal cells. * *p* < 0.001 vs. WT.

## Data Availability

The original contributions presented in the study are included in the article; further inquiries can be directed to the corresponding author.

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
