# Peer review of "Increased KV2.1 Channel Clustering Underlies the Reduction of Delayed Rectifier K+ Currents in Hippocampal Neurons of the Tg2576 Alzheimer’s Disease Mouse"

_cells, 2022, doi:10.3390/cells11182820_

Round 1

Reviewer 1 Report

The authors examines the K2.1 channel clustering in hippocampal neurons of the Tg2576 AD's disease mouse. The patch-clamp recording from primary cultured neurons, the western from cultured neurons and 3-month old were conducted. Although, the findings are very interesting, the patch-clamp recording from the primary cultured neurons were not good indication. I have a critical comments.

1. In order to support the conclusion relevant to AD condition, the patch-clamp recording should be conducted from 3-moth-old mouse both from WT and Tg2576 AD mouse. The best choice is to record Kv2.1 currents from hippocampal brain slices, you may also use acute dissociated hippocampal neuron, the data from this recording are consistent with Western or immunohistochemistry.

Author Response

Major point: 

The authors examines the K2.1 channel clustering in hippocampal neurons of the Tg2576 AD's disease mouse. The patch-clamp recording from primary cultured neurons, the western from cultured neurons and 3-month old were conducted. Although, the findings are very interesting, the patch-clamp recording from the primary cultured neurons were not good indication. I have a criticalcomments.

  1. In order to support the conclusion relevant to AD condition, the patch-clamp recording should be conducted from 3-moth-old mouse both from WT and Tg2576 AD mouse. The best choice is to record Kv2.1 currents from hippocampal brain slices, you may also use acute dissociated hippocampal neuron, the data from this recording are consistent with Western or immunohistochemistry.

Answer:

We thank the Referee for these relevant suggestions that have been done with the aim to take into consideration the gold standard of functional approaches. However, it is necessary to underline the validity of the primary cultures used in the present study to reproduce the same pathomechanisms occurring in AD transgenic mice. This is effectively demonstrated by two different groups and validated in our Lab by means of different approaches consisting on the use of either biochemical or functional techniques (Takahashi, R.H. et al. Oligomerization of Alzheimer's beta-amyloid within processes and synapses of cultured neurons and brain. J Neurosci 2004, 24:3592-9; Baldassarro, V.A. et al. Vulnerability of primary neurons derived from Tg2576 Alzheimer mice to oxygen and glucose deprivation: role of intraneuronal amyloid-β accumulation and astrocytes. Dis Model Mech, 2017, 10:671-678). In this respect, using hippocampal primary cultures at 12 DIV dissected from AD mice, we showed a perfect correlation between changes of KV2.1 protein expression measured in vitro and that studied in 3-moth-old Tg2576 mice (in the Introduction section page 2, lines 86-87; Materials and Methods section page 3, lines 120-121; Results section page 5, lines 217-219; Discussion page 14, lines 404-405 and 419-421 of the new version of the manuscript). Furthermore, new experiments on KV2.1 electrophysiological properties in hippocampal brain slices from Tg2576 mice are in progress and preliminarly confirmed the observation showed in the present study. However, due to a different scope of the in progress study, these results will converge in another future manuscript. On the other hand, the use of acute dissociated hippocampal neuron from Tg2576 AD mice is very hard and quite impossible approach in electrophysiological studies also due to the plating support on which plating dissociated neurons.

Reviewer 2 Report

Alzheimer’s disease (AD) is a neurodegenerative disease that leads to cognitive decline with age. Hyperexcitability of cortical and hippocampal neurons have been studied to be the underlying cause of dementia in AD patients. Kv2.1, a voltage-gated potassium ion channel found in these neurons has been linked neuronal hyperexcitability leading to AD pathogenesis.  Clusterization and altered function of Kv2.1 in hippocampal neuron has been studied in 3xTg animal model of AD. The work presented here by Piccialli et al., is a systematic analysis of Kv2.1 functional alteration in another AD animal model, Tg2576. The authors have performed some elegant experiments to show that Kv2.1 in Tg2576 mouse model is overexpressed but appears localized in clusters where it is nonconducting. Consequently, the authors have electrophysiological recordings to show that Kv2.1 mediated current, IDR, is reduced in these neurons affecting their excitability.

The manuscript here is well written and provides good knowledge about the role of Kv2.1 and the impairment in its electrical activity that underlies AD pathogenesis in Tg2576 mouse model.  

One minor suggestion, in Fig3B, it will be easier to follow if the first two bar graphs we were designated as WT and the next two as Tg2576 as you have done in Fig 4C.

Author Response

Minor point: 

Alzheimer’s disease (AD) is a neurodegenerative disease that leads to cognitive decline with age. Hyperexcitability of cortical and hippocampal neurons have been studied to be the underlying cause of dementia in AD patients. Kv2.1, a voltage-gated potassium ion channel found in these neurons has been linked neuronal hyperexcitability leading to AD pathogenesis. Clusterization and altered function of Kv2.1 in hippocampal neuron has been studied in 3xTg animal model of AD. The work presented here by Piccialli et al., is a systematic analysis of Kv2.1 functional alteration in another AD animal model, Tg2576. The authors have performed some elegant experiments to show that Kv2.1 in Tg2576 mouse model is overexpressed but appears localized in clusters where it is nonconducting. Consequently, the authors have electrophysiological recordings to show that Kv2.1 mediated current, IDR, is reduced in these neurons affecting their excitability.

The manuscript here is well written and provides good knowledge about the role of Kv2.1 and the impairment in its electrical activity that underlies AD pathogenesis in Tg2576 mouse model. 

One minor suggestion, in Fig3B, it will be easier to follow if the first two bar graphs we were designated as WT and the next two as Tg2576 as you have done in Fig 4C.

Answer:

We completely agree with the observation of Reviewer #2 and we modified Figure 3B

Round 2

Reviewer 1 Report

The authors explained my previous concern and did not provide additional experiment, I have no further concern.